# New Healthy Low-Sugar and Carotenoid-Enriched/High-Antioxidant Beverage: Study of Optimization and Physicochemical Properties

**DOI:** 10.3390/foods12173265

**Published:** 2023-08-30

**Authors:** Leila Abolghasemi Fakhri, Babak Ghanbarzadeh, Pasquale M. Falcone

**Affiliations:** 1Department of Food Science and Technology, Faculty of Agriculture, University of Tabriz, Tabriz P.O. Box 51666-16471, Iran; l.a.fakhri@tabrizu.ac.ir; 2Department of Agricultural, Food, and Environmental Sciences, University Polytechnical of Marche, Brecce Bianche 10, 60131 Ancona, Italy

**Keywords:** optimization, low-sugar carotenoid-enriched beverage, physicochemical characteristics, essential oil, herbal extract

## Abstract

Lutein is a prominent biologically active carotenoid pigment with a polyene skeleton that has great benefits for human health. The study examined the synergistic effects of potentially functional components, including lutein carotenoid (LC), *Mentha* × *Piperita* extract (MPE), and *Citrus* × *aurantifolia* essential oil (CAEO), all three as bioactive components and antioxidants (AOs), on the physicochemical characteristics of a new low-sugar and carotenoid-enriched high-antioxidant beverage. Sucralose was utilized as a non-nutritive sweetener. Polynomial equations obtained by combined design methodology (CDM) were fitted to the experimental data of total phenolic and flavonoid contents (TPC and TFC, respectively) and antioxidant potential of the beverages using multiple regression analysis with R^2^ (determination coefficient) values of 0.87, 0.89, and 0.97, respectively. Estimated response values for the TPC, TFC, and antioxidant potential (determined as 2, 2-diphenyl-1-picrylhydrazyl radical (DPPH^•^) scavenging activity) of the optimum beverage formulation were 41.90 mg gallic acid equivalent (GAE) per L^−1^, 27.51 mg quercetin equivalent (QE) per L^−1^, and 34.06%, respectively, with a desirability value of 0.74. The potentially functional components had a synergistic effect on the antioxidant potential. This healthy beverage can have the potential to enhance health benefits and may have therapeutic potential for diabetic patients.

## 1. Introduction

Functional beverages are one of the most active functional food categories that have become popular with consumers. These non-alcoholic beverages are enriched with nutraceuticals and bioactive components, including vitamins, phytonutrients (such as carotenoids, terpenes, and polyphenols), antioxidants, minerals, probiotics, etc., to provide health benefits beyond basic nutrition. The ready-to-drink fruit-based functional beverage sector is the fastest-growing segment in the functional food market due to ongoing new product development requirements, including convenience, “freshly prepared”, “portion control”, and “customization” [1,2]. Excessive consumption of sucrose is closely associated with sugar-sweetened beverages. This is one of the significant factors related to the increase in metabolic diseases, such as insulin resistance, obesity, and type 2 diabetes. Therefore, adding alternative sweeteners to new fruit-based beverages can help minimize the incidence or severity of these pathologies [3].

α-Carotene, β-carotene, lutein, zeaxanthin, and lycopene are the main carotenoids detected in the human body. Research involving human studies, animal models, and cell culture studies has well-demonstrated that LC has a protective role against chronic and cardiovascular diseases, light-induced skin damage, stroke, and certain types of cancer, particularly those of the breast and lung, and has a well-established role in eye health, alleviating age-related macular degeneration (AMD) and cataracts. It has also been shown that lutein has more potent anticancer activity than β-carotene against breast and prostate cancer [4,5,6]. The lutein carotenoid has been reported to have positive health effects at dietary intake levels of 6–14 mg. day^−1^ [7]. Carotenoids cannot be produced by humans. Considering that carotenoid-rich foods and supplements are the primary dietary sources of these compounds [5], the development of novel functional foods enriched with these bioactive components has gained popularity in improving the well-being and health of consumers.

Sucralose (1,6-dichloro-1,6-dideoxy-β-D-fructofuranosyl-4-chloro-4-deoxy-α-D-galactopyranoside) is one of the most frequently used sweeteners. Its popularity is due to the fact that, despite its stability, high sweetness intensity, and sugar-like sweetness, this compound is safe because it cannot be metabolized by the human body. Controlling the amount of sweeteners in food products is essential to protecting consumer health [8,9,10].

The consumption of *Citrus* × *aurantifolia* (from the Rutaceae family) juice with a high nutritional value is beneficial for human health in the scope of urinary citrate increase, cardiovascular and other degenerative disease prevention, oxidative stress relief, chronic asthma prevention, improvement in lipid profiles and inflammation markers, and neuroprotective influences, among others [1,11].

Essential oils (EOs) are aromatic and volatile liquids rich in various biologically active compounds and possess wide applications in food and pharmaceuticals owing to their diverse therapeutic activities. Citrus essential oils are the most widely used EOs in the food industry worldwide, and *Citrus* × *aurantifolia* essential oil is a valuable essential oil possessing a variety of biological activities such as antitumor, antioxidant, and antimicrobial properties [12,13,14,15].

*Mentha* × *Piperita* L. is an important aromatic and medicinal herbal plant with proven functional and therapeutic properties that can be utilized as a value-added ingredient for enrichment purposes. Its extract is an antioxidant source, blood sugar regulator, and thirst-reducer and can considerably inhibit the activity of key enzymes in hypertension and type 2 diabetes [16,17]. The peppermint leaves have physiological effects that include antioxidant, anti-inflammatory, immunomodulating, antiallergenic, antimicrobial, antitumor, and beneficial effects on the digestive system, among others [18].

To achieve their synergistic effect on the beverage’s properties, various mixtures of potentially functional components can be added simultaneously. The CDM is a useful, versatile experimental design tool for multivariate optimization design under minimal experimental points, and this technique has been suggested as a popular approach for formulation optimization purposes [19].

The production and characterization of healthy products such as low-sugar and/or potentially functional foods and beverages, as well as the investigation of their health effects, have been important areas of recent research. The formulation and characteristics of new functional low-sugar products, carotenoid-enriched oils, and fruit-based beverages have been recently evaluated [8,20,21,22,23]. Researchers have recently introduced carotenoid extract as a natural antioxidant and colorant to reduce the use of synthetic dyes, such as tartrazine, in bakery products [24]. On the other hand, studies have revealed that a functional beverage fortified with mint extract has promising antioxidant properties and is rich in total phenolic compounds [25]. It has been found that sucralose is a valuable alternative to sucrose in preventing the increased risk of several metabolic disorders [3]. There is an existing hypothesis that a diet with high antioxidant capacity could be inversely related to the development of diabetes [26].

The combination of LC, MPE, and CAEO makes sense because they can give the low-sugar lime juice-based beverage complementary properties (high physicochemical and sensory quality). All antioxidant (AO) combinations do not show synergistic effects, and the combination of some AO compounds acts antagonistically. Moreover, the impact of AOs does not usually increase with concentration, and some AOs promote oxidation at high concentrations. In addition, the effects of AOs may differ in different systems [27]. Considering what has been stated above, the current research aimed to study the synergistic or antagonistic effect of a combination of the lutein carotenoid, *Mentha* × *Piperita* extract, and *Citrus* × *aurantifolia* essential oil on physicochemical characteristics and the multi-objective optimization of these characteristics in a healthy low-sugar carotenoid-enriched lime juice-based beverage.

## 2. Materials

The *Citrus* × *aurantifolia* concentrate (∼45 °Brix), *Mentha* × *Piperita* L. extract (aqueous extract obtained from leaves of *Mentha* × *Piperita*; major components: eriocitrin and rosmarinic acid), and *Citrus* × *aurantifolia* peel essential oil (purity ≥ 99%; cold-pressed; colorless to pale-yellow oil; major components: limonene and pinene) were obtained from Takdaneh Company (Tabriz, Iran). High-methoxyl pectin (galacturonic acid, ≥74.0%; Mw 13,680 kDa) and other chemicals were of analytical reagent grade and were purchased from Sigma-Aldrich (Darmstadt, Germany).

## 3. Methods

### 3.1. Preparation of the Beverages

The *Citrus* × *aurantifolia* juice concentrate (1.85 g) was diluted with distilled water to yield a final product containing 100 g·L^−1^ of “reconstituted *Citrus* × *aurantifolia* juice with 8.3 °Brix.” Ascorbic acid (0.02% *w*/*v*) and sucralose (0.03% *w*/*v*, equivalent in sweetness to ten percent sucrose [28]) were added to the formulation, followed by stirring for 3 min. Carboxymethylcellulose and pectin were dissolved separately in distilled water (0.5% *w*/*v*) at 70 and 25 °C, respectively, for 2 h while stirring continuously. They were added to the formulation at final concentrations of 0.05% *w*/*v* [9]. The chlorophyll (0.003% *v*/*v*) was added to the mixture and stirred to disperse completely. Then the pH was adjusted to 2.5 with the addition of citric acid. This initial base formulation was then divided into 14 parts, and different beverage formulations were produced by adding potentially functional components (LC, MPE, and CAEO) depending on the combined design points (Table 1). The sensory evaluation was used to determine the high levels of potentially functional components in the formulations (Table 2), and the sum of mixture components was standardized at 5% *v*/*v* of the beverages. The beverages were then homogenized using an Ultra Turrax T25 (IKA, Staufen, Germany) at 15,000 rpm for 3 min, pasteurized at 92 ± 2 °C for 180 s, hot filled in glass bottles, cooled, and stored at 4 °C. All samples were prepared for three replications.

### 3.2. The Physicochemical Characteristics

The phytochemicals were extracted by homogenizing 2 mL of beverage samples with 10 mL of ethanol (80% *v*/*v*) by a magnetic stirrer and centrifuging the homogenates at 13,500× *g* for 15 min at 4 °C by a CR22N centrifuge (Hitachi Koki Co., Ltd., Tokyo, Japan). The supernatants were then filtered through the Whatman #1 filter paper. The extracts were stored at −20 °C to determine antioxidant potential, TFC, and TPC, as described previously [29,30]. The mean of three replications was recorded for each experiment.

#### 3.2.1. Antioxidant Potential Determined as DPPH^•^ Scavenging Activity (DPPH^•^ SA)

Briefly, 100 µL of ethanol was added to 3.9 mL of DPPH radical solution (100 µM in 80% ethanol) (blank) to assess the absorbance of the DPPH^•^ solution. Then 100 µL of specimen extract was added to 3.9 mL of 100 µM DPPH radical solution. After shaking the mixture vigorously, it was kept for 60 min in the dark at 20 °C. The decrease in absorbance was then measured at 517 nm using a UV–visible spectrophotometer (Shimadzu UV-1700, Kyoto, Japan). The percentage inhibition of the DPPH^•^ was calculated as follows:(1)$$\%\;inhibition=\frac{A_{0}-A_{s}}{A_{0}}\times 100$$
where *A_s_* is the absorbance of the sample and *A*_0_ is the absorbance of the blank.

#### 3.2.2. Total Flavonoid Content

At first, 0.2 mL of extract, 1.28 mL of deionized H_2_O, and 0.06 mL of NaNO_2_ (5%) were mixed, and after 5 min at 20 °C, 60 µL of AlCl_3_ (10% *w*/*v*) were incorporated. Six minutes later, 0.4 mL of NaOH (4% *w*/*v*) was incorporated under the same conditions. After stirring, the absorbance was assessed at 510 nm by a UV–visible spectrophotometer (Shimadzu UV-1700, Japan). The calibration curve of the quercetin was made up in the range of 10–200 mg·L^−1^. The TFC was expressed as mg of quercetin equivalents (QE)·L^−1^ of the specimen.

#### 3.2.3. Total Phenolic Content

In brief, 200 μL of diluted extracts (1:20) were incorporated into 1 mL of the 10% *v/v* Folin–Ciocalteu reagent and incubated for 3 min at 20 °C. Then 0.8 mL of Na_2_CO_3_ (75 g·L^−1^) was added. After 2 h of incubation of the mixture at 20 °C, the absorbance was measured at 765 nm by a UV–visible spectrophotometer (Shimadzu UV-1700, Japan). The content of total phenolics was measured using gallic acid as a standard. The standard curve of gallic acid was made up in the range of 10–200 mg·L^−1^. Results were expressed as mg gallic acid equivalents (GAE)·L^−1^ of the specimen.

#### 3.2.4. Titratable Acidity (TA), Total Soluble Solids (TSS), and pH

The TSS (°Brix) and pH were determined by a Mettler Toledo (Greifensee, Switzerland) refractometer and pH meter, respectively, at 20 °C. The TA (% citric acid) was measured by titrating a 10 mL specimen with NaOH (0.1 N) using phenolphthalein till the specimen turned light pink (pH 8.1) [31] and was calculated as follows:(2)$$\%\;titratable\;acidity=\frac{0.064\times 100\times V\times 0.1N\;NaOH}{V_{s}}$$
where *Vs* is the volume (mL) of the specimen and *V* (mL) is the titer volume of NaOH. Tests were carried out in triplicate.

#### 3.2.5. Ascorbic Acid Content (AA)

The iodine titration method was used to quantify the amount of ascorbic acid as described, considering that each mL of the iodine solution is equivalent to 0.88 mg of AA [32]. In brief, 20 mL of the specimen was added to 150 mL of distilled water and titrated with an iodine solution till a stable dark blue color was obtained. The starch solution (10 g·L^−1^) was used as the indicator. The following equation was used to estimate the ascorbic acid content:(3)mg ascorbic acid/100 mg sample=0.88×mL iodine solution

The iodine solution was prepared as follows: potassium iodate (KIO_3_) (0.268 g) and potassium iodide (KI) (5 g) were dissolved in 200 mL of distilled water. Sulfuric acid (30 mL, 3 M) was added, and the volume of the solution was adjusted to 500 mL with distilled water. Tests were carried out in triplicate.

### 3.3. Determination of Levels of LC, MPE, and CAEO to Be Incorporated into Beverages by Using a Sensory Panel

The high level of LC, MPE, and CAEO (diluted 1:10) for the formulation of beverages was determined based on the sensory evaluation, which was carried out by a panel comprising 30 semi-trained members (16 females and 14 males) involving students of Tabriz University, Department of Food Science and Technology, Tabriz, Iran. The age of the panelists was between 23 and 40 years old. The panelists assessed the beverage samples for taste, odor, color, and overall acceptance attributes on a 9-point hedonic scale, with “dislike extremely” scored as 1, “dislike very much” scored as 2, “dislike moderately” scored as 3, “dislike slightly” scored as 4, “neither like nor dislike” scored as 5, “like slightly” scored as 6, “like moderately” scored as 7, “like very much” scored as 8, and “like extremely” scored as 9 [20]. The panelists received transparent polyethylene containers with sealable lids containing approximately 40 mL of the beverage specimen at 4 °C, coded with three random digits and a random order, with salt-free crackers and drinking water between samples. The experimental protocols were approved by the Research Ethics Board of the University of Tabriz, and the evaluation took place in the sensory evaluation booths under white artificial light. The tests were carried out in three replications.

### 3.4. Design of Experiments and Statistic Methods

The CDM-D-optimal with independent variables, including one mixture factor (CAEO/MPE ratio) and one numeric factor (LC concentration at five levels ranging between 0.001 and 0.003% *w*/*v*), was used to find out the influence of the independent variables and their possible interactions, to model the physicochemical characteristics, including in vitro antioxidant potential, TFC, and TPC (dependent variables), as a function of the independent variables, and finally to estimate the optimum beverage formulation considering the physicochemical characteristics. Fourteen experimental runs were obtained by the design, as presented in Table 2. Design-Expert software (Stat-Ease Inc., Minneapolis, MN, USA) was applied to construct the design and perform all the data analysis. The contour and 3D surface plots were also obtained with this software. The following polynomial equation was fitted to the data:(4)$$Y=\beta_{0}+\sum_{i=1}^{k}\beta_{i}X_{i}+\sum_{i=1}^{k}\beta_{ii}X_{i}^{2}+\sum_{\begin{array}{c} i=1\\ i<j \end{array}}^{k-1}\sum_{j=2}^{k}\beta_{ij}X_{i}X_{j}$$
where *β*_0_, *Y*, and *k* correspond to a constant coefficient, a response variable, and the number of variables, respectively. *β_ii_*, *β_ij_*, and *β_i_* are the measures of the $X_{i}^{2}$, *X_i_X_j_*, and *X_i_* of quadratic, interaction, and linear effects, respectively. *X_j_* and *X_i_* are independent variables in coded units.

Statistically significant differences between means were calculated by the one-way analysis of variance (ANOVA) and Duncan’s multiple range test at the 0.05 significance level by SPSS software (SPSS Inc., Chicago, IL, USA).

### 3.5. Overall Optimization Procedure

The maximum values of DPPH^•^ SA, TFC, and TPC were determined by numerical and graphical optimization techniques using Design-Expert software (version 10).

### 3.6. Verification Tests and Validation of the Models

The equations’ adequacy was checked by verification tests using the optimum levels of factors. The optimized formulation was developed using the same procedure as described earlier. The optimum beverage specimen and its replication were compared regarding their physicochemical properties. The validity of the models obtained was evaluated by comparing the actual data to those estimated.

## 4. Results

### 4.1. Determination of Levels of LC, MPE, and CAEO to Be Incorporated into Beverages by Using a Sensory Panel

The sensory changes caused by LC, MPE, and CAEO were notably appreciated. Incorporating LC, MPE, and CAEO into the beverage formulation has an enhanced effect on the sensory attributes of the beverage.

The sample without additives was the least accepted product, with the minimum mean sensory acceptance scores (*p* < 0.05). The panelists’ preferences in terms of sensory characteristics (taste, odor, color, and overall acceptance) increased with increasing LC, MPE, and CAEO content to a certain extent. Still, it decreased with further increases (*p* < 0.05). (Table 1). Considering the mean sensory score for taste, odor, and overall acceptance, there was no significant difference between the four levels (2, 3, 4, and 5% *v*/*v*) of MPE and CAEO (*p* < 0.05). Increasing the content of MPE and CAEO caused a significant decrease in the mean sensory scores (*p* < 0.05). On the other hand, no significant difference was observed between different levels of LC in terms of taste, and the sample containing 0.004% *w*/*v* LC was the product with a mean color and overall acceptance scores lower than the sample containing 0.003% *w*/*v* LC (*p* < 0.05). Therefore, the high levels of LC, MPE, and CAEO determined to be incorporated into the beverage formulation were 0.003% *w*/*v* LC and 5% (*v*/*v*) MPE and CAEO (diluted 1:10).

### 4.2. Optimization of the Physicochemical Characteristics

#### 4.2.1. Regression Models Analysis

The design of experiments, levels of factors, and actual and estimated data for the dependent variables are indicated in Table 2. Beverage formulation affected the DPPH^•^ SA, TFC, and TPC. To explain the influence of the components on the response variables and generate the equations, various models were fitted to the data. The models’ adequacy was evaluated with three statistical parameters, including lack of fit tests (LOF), model summary statistics (MSS), and sequential *p*-values (Table 3). Considering the insignificant LOF (*p* > 0.05), significant sequential *p*-value (*p* < 0.01), and highest determination coefficient (R^2^), adjusted R^2^, and predicted R^2^, the linear × mean model was chosen as the most proper model for total phenolic and flavonoid contents. The quadratic × linear model was chosen for DPPH^•^ SA (Table 3).

The non-significant lack of fit (*p* > 0.05) and highly significant model (*p* < 0.001) indicated the adequacy of selected models for responses (Table 4). The developed models showed good predictive success with determination coefficient (R-squared) values above 0.85, which indicated the good fit of the models and their ability to describe the influence of the potentially functional components on the physicochemical characteristics (Table 4).

The models’ general availability and adequate accuracy were demonstrated by a reasonable agreement between the adjusted R^2^ and the predicted R^2^ for responses. The C.V. (coefficient of variation) values < 10% provided better reproducibility and showed high accuracy and good reliability of the true values. The three models have adequate precisions that are greater than 4, which indicates adequate signals [33]. Therefore, the models created are suitable for analyzing the responses.

The linear effects of both mixture components (*X*_1_ and *X*_2_) were the most significant terms affecting all three models (*p* < 0.01). For DPPH^•^ SA, the *X*_1_*X*_2_ and *X*_1_*X*_3_ were also significant model parameters (*p* < 0.01) (Table 4). When considering the regression coefficients obtained for variables, the MPE (*X*_2_) concentration was found to be the most influential parameter affecting the TPC and TFC, and the interaction effect of mixture components (CAEO and MPE) was the most critical factor affecting antioxidant potential (Table 4 and Equations (5) to (7)). The final equations in terms of coded factors after excluding the non-significant coefficients were as follows:(5)$$Y_{1}=23.74X_{1}+32.51X_{2}+39.56X_{1}X_{2}+2.58X_{1}X_{3}$$
(6)$$Y_{2}=16.81\;X_{1}+27.63\;X_{2}$$
(7)$$Y_{3}=31.60\;X_{1}+42.01\;X_{2}$$

*Y*_1_, *Y*_2_, and *Y*_3_ are DPPH^•^ SA, TFC, and TPC, respectively.

#### 4.2.2. Analysis of Response Surface

##### The Combined Effects of Potentially Functional Components on DPPH^•^ SA

The convex surface indicated a quadratic effect of the CAEO/MPE combination on DPPH^•^ SA (Figure 1a and Figure 2a). By increasing the amount of MPE in the mixture, first an upward trend and then a downward trend were observed for the antioxidant potential of the beverages. A specimen containing an approximately 38: 62 CAEO: MPE mixture combination showed, in general, the highest DPPH^•^ SA at all LC amounts (*p* < 0.05). At the same mixture component amount, the DPPH^•^ SA of beverages increased linearity with increasing LC content, and this increase was more significant in the specimens containing only CAEO (Figure 1a and Figure 3a). The CAEO and LC had a positive interaction and synergistic effect on antioxidant activity (Figure 1a and Figure 3a, and Equation (5)). The specimen containing 0.001% *w*/*v* LC and 100% CAEO in the mixture had minimum antioxidant activity.

##### The Combined Effects of Potentially Functional Components on Total Flavonoid Content (TFC)

Essential oil/extract combination linearly affected the TFC. Total flavonoid content values increased linearly with increasing MPE in the mixture (*p* < 0.05) (Figure 1b, Figure 2b and Figure 3b, and Equeation (6)). The TFC value behaved almost identically at all LC levels. The LC amount did not affect the TFC (Figure 2b and Figure 3b). The highest value of TFC was obtained using 100% MPE in the mixture.

##### The Combined Effects of Potentially Functional Components on Total Phenolic Content (TPC)

Essential oil/extract combination demonstrated a linear impact on TPC value, and the LC level did not affect this parameter (*p* < 0.05) (Figure 1c, Figure 2c and Figure 3c, and Equation (7)). The maximum TPC value was obtained in the specimens containing only MPE.

#### 4.2.3. Overall Optimization Procedure

A combination of the optimal level of independent variables with maximum desirability was achieved using the desirability function technique. For multi-response optimization, the desired goal for independent variables and responses was set to “within the range” and “maximum,” respectively, and an importance value of 5 was assigned for all responses. The selected weighting factor was 1. Only one solution was obtained (Table 5). Figure 4a shows the desirability ramps generated from the optimal points through optimization. Figure 4b shows the maximum overall desirability and the individual desirability for each dependent variable at the optimum point. The maximum desirability obtained (0.74) corresponded to “good desirability” (>0.63) and indicated that the design was suitable for use.

The AA content, total acidity, pH, and TSS of the beverage produced with the optimum formula are presented in Table 5.

#### 4.2.4. Verification Tests and Validation of the Models

Table 5 demonstrates the true and estimated amounts of the dependent variables at the optimum point and the percentage of error between them. The closeness of the true values to the estimated ones and an acceptable percentage error (less than 30%) [34] showed the adequacy and validity of the models and optimization process.

## 5. Discussion

Our findings are consistent with previous studies showing the improving effects of lutein [35,36], herbal extracts such as mint extract [25,37], and essential oils [38] on the physicochemical properties of fortified/functional beverages and foods. Studies showed an increase in DPPH scavenging activity and TPC of carotenoid-enriched cottonseed oil compared to the corresponding values for cottonseed oil [23]. Improving the chromatic and flavor properties and increasing the oxidative stability of soy milk mayonnaise using carotenoid-enriched oils have been reported. An increase in oxidative stability and improved sensory characteristics of carotenoid-enriched mayonnaise have been observed with the addition of basil essential oil [38]. Research has shown that lutein preserves the quality of yogurt during its storage period by protecting riboflavin from oxidation [36]. The synergistic effects of ascorbyl palmitate, vitamin E, and rosemary extract on the antioxidant potential of microalgal docosahexaenoic acid-rich oil have been reported [27]. A little synergistic effect between neutral phenols and AA on antioxidant potential and an antagonistic effect between these bioactive compounds with increasing AA concentration have been reported [39]. In another study, the antagonistic action of the combination of phenols derived from plants and alpha-tocopherol sunflower seed oil was reported [40]. This antagonistic action has been attributed to the selectivity of the microenvironment between AOs or the competition in the free radical formation of various active AOs. No previous research has optimized LC, MPE, and CAEO concentrations simultaneously to assess possible synergistic effects of them on product physicochemical properties.

The special ability of LC to quench singlet oxygen and other reactive oxygen species (ROS) makes it a powerful antioxidant. There are several studies on the antioxidant potential of LC (Figure 5a) and carotenoid-enriched oil [4,23,41]. The *Citrus* × *aurantifolia* fruit is an important source of bioactive compounds and nutrients, including essential minerals, citric acid, ascorbic acid and other vitamins, flavonoids and other phenolic compounds (PC), and dietary fiber [42]. Citrus flavonoids (flavanones, flavones, and flavonols) are particular nutrients that are rare in other types of fruit, and flavanone compounds form the major part of citrus phenolics [43]. The main flavanones, flavanone aglycones, and glycosides in lime are rutinose, neohesperidose, hesperetin, naringenin, isosakuranetin, eriodictyol, diosmetin, eriocitrin, narirutin, hesperidin, diosmin, and neoeriocitrin (Figure 5b). Hesperidin is widely distributed among citrus fruits and is the primary flavanone in lime. Eriocitrin is in second place after hesperidin. Lime and lemon are the only citrus fruits that contain a significant amount of this compound. The phenolic compounds of citrus juices are one of the important contributors to their antioxidant capacity. Eriocitrin is an excellent bioceutical with a higher antioxidant activity than the other flavonoid compounds present in citrus. Ascorbic acid, an efficient ROS scavenger, is another crucial antioxidant in citrus juices. Although eriocitrin is heat stable, ascorbic acid is heat labile and very sensitive to different processing conditions as well as to light [44,45].

Almost all the EOs are exceptionally rich in secondary metabolites (terpenes, terpenoids, flavonoids, nitrogen- and sulfur-containing alkaloids, and other aromatic and aliphatic constituents). Still, their chemical composition can vary markedly depending on the specific oil, species, season of harvesting, and geographical origin [46]. They consist mainly of terpenic compounds and usually contain fewer polyphenols than plant extracts. *Citrus* × *aurantifolia* essential oil is a complex mixture of organic compounds with potential biological functions, such as limonene, γ-terpinene, citral, linalool, and β-caryophyllene, among others, which may be represented by three main classes: terpenes (75%), oxygenated compounds (12%), and sesquiterpenes (3%) [12]. Limonene (a monoterpene) constitutes the most significant volatile, followed by γ-terpinene, β-pinene, α-pinene, neryl acetate, and sabinene [47]. In addition, cold-pressed CAEO contains non-volatile chemicals (~20%), which are mainly coumarin and psoralen derivatives, as well as colorants and wax. Bergamottin and 5-geranyloxy-7-methoxy coumarin are the most abundant oxygen heterocyclic compounds identified in cold-pressed CAEO [42,44] (Figure 5c). *Citrus* × *aurantifolia* essential oil has been found to have an intense antioxidant capacity. The DPPH^•^ SA of the CAEO at a concentration range of 80–3460 mg·L^−1^ was reported as 10.65–66.44% [47]. Terpenes, flavonoids, carotenes, and coumarins are responsible for the intense antioxidative activities of citrus essential oils [44].

*Mentha* × *Piperita* extract contains large amounts of flavonoid and phenolic antioxidants and lower amounts of terpenes and vitamins. Phenolic acids (42% *w*/*w*), flavonoids (53% *w*/*w*), lignans, and stilbenes (2.5% *w*/*w*) make up the major part of its polyphenolic content (0.75 g·L^−1^). Eriocitrin, luteolin 7-O-rutinoside, rosmarinic acid, and eriodictyol-glucopyranosyl-rhamnopyranoside are the most abundant phenolics (Figure 5d) [16,17] Flavanones (e.g., eriodictyol glycosides), flavones (e.g., luteolin glycosides), and phenolic acids (e.g., caffeic and rosmarinic acids) are likely the major antioxidants of MPE, and vitamins (for instance, AA) have a small contribution to its overall antioxidant activity [48]. The high phenolic content of MPE could be the main reason for its high antioxidant activity towards DPPH free radicals (FRs), and researchers have reported a direct positive correlation between its DPPH radical-scavenging activity and TPC [49].

Depending on their mechanism of action, AOs typically fall into the following categories [50,51]: 1—AOs that can directly scavenge FRs or delay the initial formation of free radical species; 2—Those that block or slow down autoxidation through competing with the propagation reactions (these compounds are called chain-breaking AOs); 3—AOs that reduce the overall oxidation rate by shortening the chain length (these compounds are called termination-enhancing AOs); 4—compounds that counteract pro-oxidant metals (chelators); and 5—Those that can increase AO defense in living systems [48]. Phenolic compounds can act as free radical scavengers, hydrogen- or electron-donating agents, singlet oxygen quenchers, and metal chelators, which makes them good antioxidants [52,53]. The literature has well-documented the structure-action relationship in each class of phenolic compounds [51]. The meta 5,7-dihydroxy arrangements in ring A, the ortho 3′,4′-dihydroxy moiety in ring B, and the 2,3-hydroxy group in ring C are the most relevant structural factors for the delocalization of electrons. Terpenes can act as antioxidants via the direct ROS scavenging pathway. Unsaturated terpenes with a cyclohexadiene structure are classified as termination-enhancing antioxidants that reduce the overall oxidation rate. Carotenoids (tetraterpenes), such as lutein, have been introduced as important antioxidants due to the multiple bonds in their structure. Moreover, lutein can show high scavenging potential due to the length of its conjugated system and the –OH groups in the terminal ring’s double bonds [48,50].

According to the results, the developed low-sugar carotenoid-enriched beverage contains an acceptable amount of health-promoting compounds. This mix of antioxidant phytochemicals (LC, AA, PC, terpenes, etc.) can help the body protect against the oxidative activity of ROS and has the potential to prevent the biological structure damage and development of many human diseases that occur in pathological situations where there is a potential redox imbalance and the endogenous AO defense systems (for example, catalase and glutathione) cannot eliminate excessive amounts of reactive oxygen species [54,55]. Inflammation- and oxidative stress-related disorders, such as chronic, cardiovascular, neurodegenerative, neurological, and diabetic disorders, and cancers, all show robust evidence of ROS involvement [56]. Adopting a bioactive-rich diet, such as one with high antioxidant capacity, has been suggested to improve human health by maintaining the body’s main functions and preventing diseases. The therapeutic roles of phytochemicals in health promotion and disease prevention have been extensively reviewed using various activities such as antioxidant, antitumor, anticancer, antidiabetic, antiobesity, antiviral, antimicrobial, antianxiety, neuroprotection, hepatoprotection, and immunomodulation. Adopting a bioactive-rich diet is also an effective strategy for reversing central adiposity and metabolic and oxidative stress. The effective action of antioxidants on the global network of oxidative stress may have a significant effect on aging and related diseases [57,58]. However, the safety and health risks of natural antioxidants require further study so that optimal levels of these substances can be safely incorporated into foods without compromising nutritional and sensory properties. Although using several substances that can lead to synergistic effects at lower levels can be an alternative strategy, further study regarding the side effects and toxicity of antioxidant phytochemicals is highly needed [59,60].

## 6. Conclusions

The bioactive compound content in terms of the TPC, TFC, and antioxidant potential (the DPPH^•^ SA) of new nutraceutical low-sugar beverages enriched with LC, CAEO, and MPE was successfully optimized by applying the CDM. The synergistic effect of potentially functional components on antioxidant potential was confirmed. The analysis of variance showed that the equations created were well-fitted with the true data. The overall optimum bioactive compound content and antioxidant potential were estimated to be achieved using a 0.05: 4.95 (*v*/*v*) ratio of CAEO: MPE and 0.003% *w*/*v* LC, with “good desirability.” The closeness of the true values to the estimated ones confirmed the adequacy and validity of the optimization process. The AA content, TA, pH, and TSS of the final optimal beverage were 25.76 mg/100 g^−1^, 0.59% citric acid, 2.55, and 1 °Brix, respectively. This healthy sugar-free beverage with acceptable and promising sensorial, physicochemical, potential bioactive components, and antioxidant traits can have potential applications in enhancing health benefits and therapeutic purposes and thus be attractive for the growing market of high nutritional value and health-promoting food products. Some in vitro (gastrointestinal simulations to prove the bioavailability) or even in vivo (clinical) studies are needed in the near future to claim that this healthy sugar-free beverage is functional. After performing the mentioned experiments, this developed beverage can have potential applications and commercialization as a functional beverage. Thus, in vitro or in vivo studies are a potentially important field for future studies.

## Figures and Tables

**Figure 1 foods-12-03265-f001:**
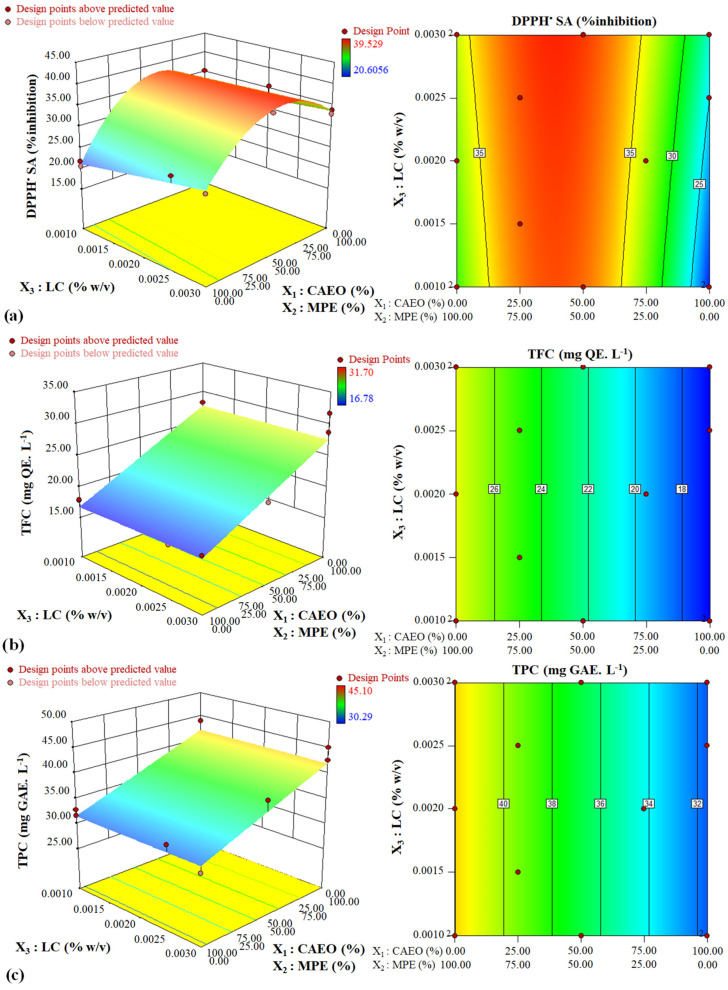
Changes in the (**a**) antioxidant capacity, (**b**) TFC, and (**c**) TPC of beverages at different levels of LC and different CAEO: MPE ratios (in color).

**Figure 2 foods-12-03265-f002:**
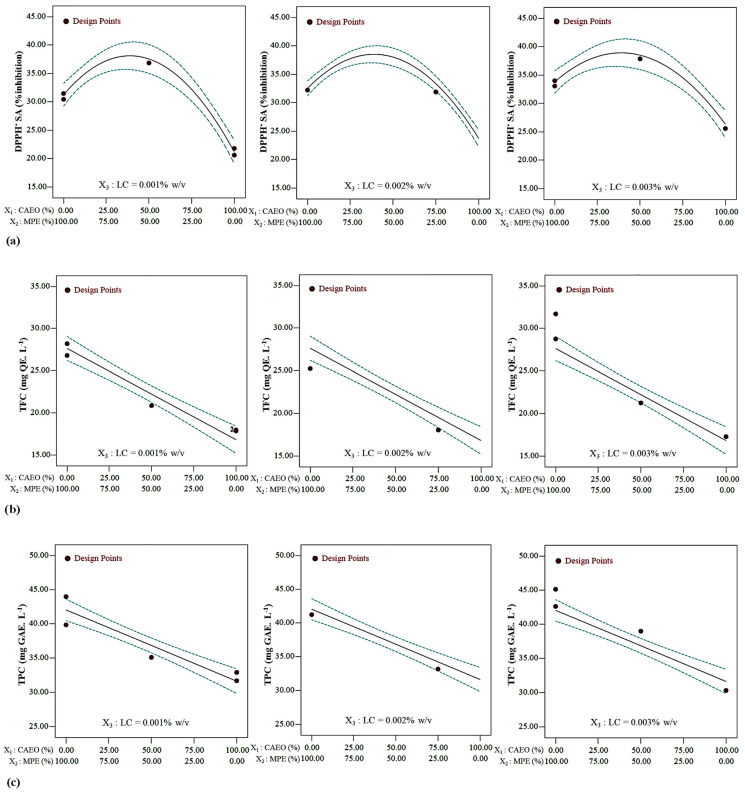
Changes in the (**a**) antioxidant capacity, (**b**) TFC, and (**c**) TPC of beverages at specified LC content and different CAEO: MPE ratios (in color).

**Figure 3 foods-12-03265-f003:**
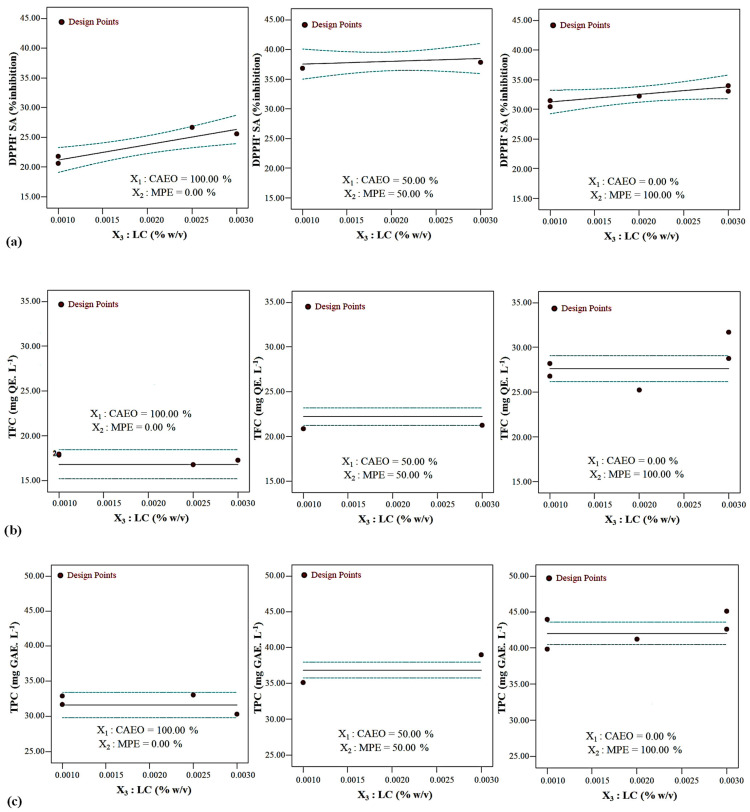
Changes in the (**a**) antioxidant capacity, (**b**) TFC, and (**c**) TPC of beverages at different content of LC and constant CAEO: MPE ratio (in color).

**Figure 4 foods-12-03265-f004:**
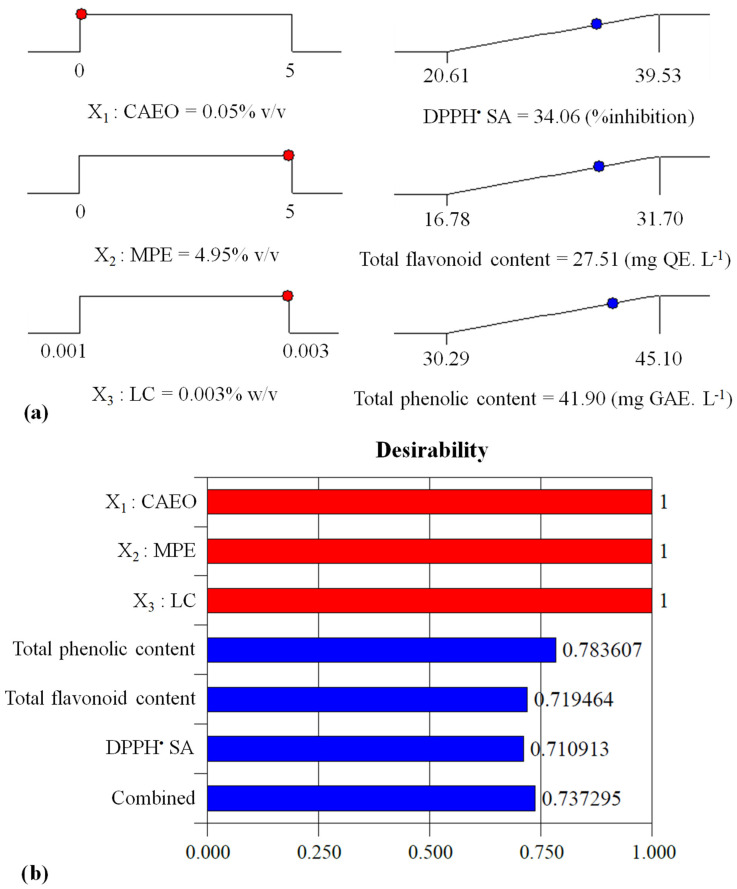
(**a**) Desirability ramp for optimization of the antioxidant capacity and total flavonoid and phenolic contents of beverages; and (**b**) the individual desirability for each dependent variable at the optimal levels of independent variables (in color).

**Figure 5 foods-12-03265-f005:**
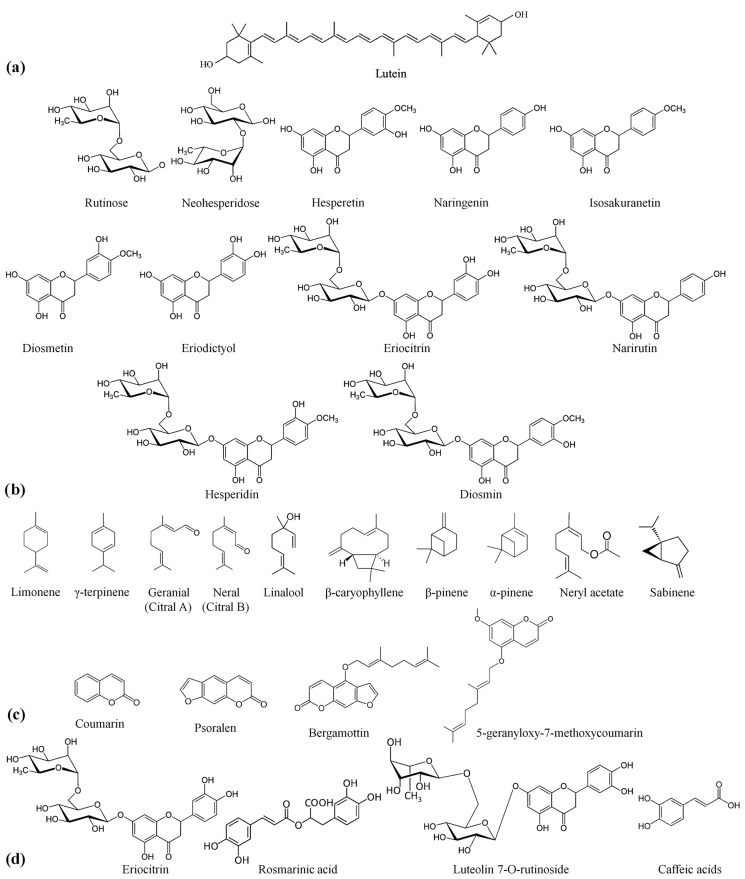
Chemical structures of (**a**) lutein, (**b**) the main flavanones, flavanone aglycones, and glycosides in lime, (**c**) the main terpenes, oxygenated compounds, and sesquiterpenes of CAEO, and (**d**) the most abundant phenolic compounds of MPE (black and white).

**Table 1 foods-12-03265-t001:** Design of experiment for optimizing three potentially functional components of healthy beverages and the responses, the factors levels, and the true and estimated data for the responses.

Design Point	Components		Factors	Responses ^e^					
	Mixture Components		Numeric Factor	DPPH^•^ SA ^f^ (% Inhibition)		Total Flavonoid Content (mg QE ^g^·L^−1^)		Total Phenolic Content (mg GAE ^h^·L^−1^)	
	*X*_1_: CAEO ^a^ (Diluted 1:10) (% *v*/*v*) ^b^	*X*_2_: MPE ^c^ (% *v*/*v*) ^a^	*X*_3_: LC ^d^ (% *w*/*v*)	Actual Value	Estimated Value	Actual Value	Estimated Value	Actual Value	Estimated Value
1	0.00	5.00	0.0020	32.21 ± 0.64	32.51	25.25 ± 0.76	27.63	41.20 ± 2.47	42.01
2	2.50	2.50	0.0010	36.84 ± 1.84	37.54	20.86 ± 1.04	22.22	35.07 ± 1.75	36.80
3	5.00	0.00	0.0010	21.78 ± 1.09	21.16	17.85 ± 0.71	16.81	32.88 ± 2.30	31.60
4	5.00	0.00	0.0030	25.57 ± 0.51	26.32	17.27 ± 1.21	16.81	30.29 ± 1.21	31.60
5	1.25	3.75	0.0015	39.19 ± 1.57	37.48	23.96 ± 1.44	24.93	36.85 ± 1.84	39.40
6	5.00	0.00	0.0025	26.66 ± 0.27	25.03	16.78 ± 1.01	16.81	33.02 ± 0.66	31.60
7	2.50	2.50	0.0030	37.85 ± 0.38	38.49	21.25 ± 0.64	22.22	38.94 ± 1.17	36.80
8	3.75	1.25	0.0020	31.88 ± 0.96	33.35	18.04 ± 0.36	19.52	33.14 ± 2.32	34.20
9	0.00	5.00	0.0010	31.46 ± 0.63	31.24	28.19 ± 1.13	27.63	39.82 ± 1.59	42.01
10	1.25	3.75	0.0025	39.53 ± 1.58	37.99	24.57 ± 1.97	24.93	38.53 ± 1.93	39.40
11	0.00	5.00	0.0010	30.45 ± 1.83	31.24	26.79 ± 1.61	27.63	43.96 ± 2.64	42.01
12	5.00	0.00	0.0010	20.61 ± 0.82	21.16	17.96 ± 0.90	16.81	31.66 ± 0.63	31.60
13	0.00	5.00	0.0030	33.98 ± 1.02	33.78	31.70 ± 0.63	27.63	45.10 ± 1.80	42.01
14	0.00	5.00	0.0030	33.05 ± 1.98	33.78	28.75 ± 1.15	27.63	42.59 ± 2.56	42.01

^a^ *Citrus* × *aurantifolia* essential oil; ^b^ ml/100 mL of formulation; ^c^ *Mentha* × *Piperita* extract; ^d^ Lutein carotenoid; ^e^ Data are expressed as means ± standard deviation (*n* = 3); ^f^ DPPH^•^ scavenging activity; ^g^ Quercetin equivalent; ^h^ Gallic acid equivalent.

**Table 2 foods-12-03265-t002:** Analysis of taste, odor, color, and overall acceptance of beverages with different levels of LC, MPE, and CAEO.

	Sensory Scores ^d^		
Levels of LC ^a^ (% *w*/*v*)	Taste	Color	Overall Acceptance
0.0	4.4 ± 1.4 ^a^	4.1 ± 1.2 ^c^	4.4 ± 1.4 ^c^
0.001	4.5 ± 1.1 ^a^	5.8 ± 0.8 ^a^	5.2 ± 1.1 ^a,b^
0.002	4.6 ± 1.1 ^a^	5.9 ± 0.5 ^a^	5.4 ± 1.1 ^a^
0.003	4.8 ± 1.3 ^a^	6.2 ± 0.8 ^a^	5.6 ± 1.1 ^a^
0.004	4.8 ± 1.3 ^a^	4.7 ± 1.1 ^b^	4.8 ± 1.2 ^b,c^
Levels of MPE ^b^ (% *v*/*v*)	Taste	Odor	Overall acceptance
0.0	4.5 ± 1.3 ^c^	6.3 ± 1.2 ^b^	5.0 ± 1.8 ^c^
2.0	6.3 ± 1.3 ^a,b^	6.9 ± 1.2 ^a^	6.7 ± 1.2 ^a^
3.0	6.7 ± 1.4 ^a^	7.2 ± 1.1 ^a^	7.0 ± 1.3 ^a^
4.0	6.7 ± 1.5 ^a^	7.0 ± 1.3 ^a^	6.9 ± 1.3 ^a^
5.0	6.5 ± 1.7 ^a^	6.9 ± 1.3 ^a^	6.7 ± 1.5 ^a^
6.0	5.9 ± 1.7 ^b^	6.1 ± 1.3 ^b^	6.1 ± 1.4 ^b^
Levels of CAEO ^c^ (diluted 1:10) (% *v*/*v*)	Taste	Odor	Overall acceptance
0.0	4.5 ± 1.2 ^c^	6.4 ± 1.3 ^c^	4.8 ± 1.3 ^c^
2.0	6.7 ± 1.4 ^a^	7.3 ± 1.1 ^a^	7.1 ± 1.2 ^a^
3.0	6.9 ± 1.4 ^a^	7.4 ± 0.8 ^a^	7.3 ± 1.1 ^a^
4.0	7.1 ± 1.1 ^a^	7.5 ± 0.8 ^a^	7.4 ± 1.0 ^a^
5.0	7.0 ± 1.2 ^a^	7.3 ± 0.8 ^a^	7.2 ± 1.0 ^a^
6.0	6.1 ± 1.4 ^b^	6.8 ± 0.7 ^b^	6.4 ± 1.1 ^b^

^a^ Lutein carotenoid; ^b^ *Mentha* × *Piperita* extract; ^c^ *Citrus* × *aurantifolia* essential oil; ^d^ Means of (liking scores) hedonic rating (1–9) for sensory attributes; Values are presented as mean ± standard deviation, *n* = 3; A higher value is more acceptable; Data with different superscripts in each column are significantly different (*p* < 0.05).

**Table 3 foods-12-03265-t003:** ANOVA and model fit summary.

Source	Suggested Models		Partial Sum of Squares		Lack of Fit (LOF)	Model Summary Statistics (MSS)			Sequential *p*-Value	
	Mix Order	Process Order	Sum of Squares	Mean Square		R^2^	Adjusted R^2^	Predicted R^2^	Mix	Process
DPPH^•^ SA ^a^	Quadratic	Linear	460.82	92.16	0.1257	0.9713	0.9534	0.9141	<0.0001 **	0.0235 *
Total flavonoid content	Linear	Mean	280.50	280.50	0.3548	0.8943	0.8855	0.8546	<0.0001 **	-
Total phenolic content	Linear	Mean	259.65	259.65	0.6764	0.8650	0.8538	0.8195	<0.0001 **	-

^a^ DPPH^•^ scavenging activity; **: Significant at a *p*-level < 0.01. *: Significant at a *p*-level < 0.05.

**Table 4 foods-12-03265-t004:** Regression coefficients and variance and regression analysis.

Source	DPPH^•^ SA ^b^			Total Flavonoid Content			Total Phenolic Content		
	Reg. Co.^a^	F-Value	*p*-Value	Reg. Co.	F-Value	*p*-Value	Reg. Co.	F-Value	*p*-Value
*X* _1_	23.74	-	-	16.81	-	-	31.60	-	-
X2	32.51	-	-	27.63	-	-	42.01	-	-
X1X2	39.56	137.45	<0.0001 **	-	-	-	-	-	-
X1X3	2.58	12.53	0.0076 **	-	-	-	-	-	-
X2X3	1.27	3.87	0.0848	-	-	-	-	-	-
X1X2X3	−5.81	2.11	0.1843	-	-	-			
Model	-	54.22	<0.0001 **	-	101.56	<0.0001 **	-	76.89	<0.0001 **
Linear mixture	-	111.51	<0.0001 **	-	101.56	<0.0001 **	-	76.89	<0.0001 **
Lack of fit	-	4.41	0.1257	-	1.74	0.3548	-	0.75	0.6764
Determination coefficient (R^2^)	0.9713	-	-	0.8943	-	-	0.8650	-	-
Adjusted R^2^	0.9534	-	-	0.8855	-	-	0.8538	-	-
Predicted R^2^	0.9141	-	-	0.8546	-	-	0.8195	-	-
Adequate precision	20.305	-	-	17.220	-	-	14.984	-	-
Coefficient of variation %	4.14	-	-	7.29	-	-	4.92	-	-
Standard deviation	1.30	-	-	1.66	-	-	1.84	-	-
PRESS	40.76	-	-	45.62	-	-	54.17	-	-

^a^ Regression coefficient; ^b^ DPPH^•^ scavenging activity; **: Significant at a *p*-level < 0.01.

**Table 5 foods-12-03265-t005:** The optimum point, desirability value, the true and estimated values of responses at the optimum point, the percentage error between the true and estimated data, and the physicochemical properties of the beverage prepared with the optimum formulation.

Optimum Formula				Responses at Optimum Point			
Mixture Components		Numeric Factor					
CAEO ^a^ (Diluted 1:10) (*v*/*v*) ^a^	MPE ^b^ (*v*/*v*) ^c^	LC ^d^ (% *w*/*v*)	Desirability		DPPH^•^ SA ^e^ (% Inhibition)	Total Flavonoid Content (mg QE ^f^·L^−1^)	Total Phenolic Content (mg GAE ^g^·L^−1^)
0.05	4.95	0.003	0.737	EV ^h^	34.060	27.514	41.895
				TV ^i^	40.15 ± 0.95	24.84 ± 1.19	51.79 ± 1.80
				PE ^j^ (%)	15.18	−10.76	19.10
Physicochemical properties ^k^							
Total soluble solids (°Brix)		pH		Ascorbic acid ^l^		Titratable acidity ^m^	
1.00 ± 0.0		2.55 ± 0.04		25.76 ± 1.67		0.59 ± 0.03	

^a^ *Citrus* × *aurantifolia* essential oil; ^b^ *Mentha* × *Piperita* extract; ^c^ mL/100 mL of formulation; ^d^ Lutein carotenoid; ^e^ DPPH^•^ scavenging activity; ^f^ Quercetin equivalent; ^g^ Gallic acid equivalent; ^h^ Estimated value; ^i^ True value (mean ± standard deviation, *n* = 3); ^j^ Percentage error; ^k^ Values are presented as mean ± standard deviation, *n* = 3; ^l^ mg· 100 g^−1^; ^m^ % citric acid.

## Data Availability

Data is contained within the article.

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
