# Peer review of "New Healthy Low-Sugar and Carotenoid-Enriched/High-Antioxidant Beverage: Study of Optimization and Physicochemical Properties"

_foods, 2023, doi:10.3390/foods12173265_

Round 1
Reviewer 1 Report
The study is interesting, but it presents several errors that are important to address.
These errors are highlighted in the attached file.
According to the plagiarism analysis, 32% is detected. It is recommended to work on the manuscript to reduce this value.
There is a lack of discussion in the document, only results are discussed but there is no comparison or discussion with other works.

Author Response
Reviewer 1
Comments and Suggestions for Authors
The study is interesting, but it presents several errors that are important to address.
We appreciate the valuable comments from the dear reviewer and his or her positive feedback. We considered these comments, and we revised our manuscript according to the comments as follows:
These errors are highlighted in the attached file.
Answer:
We appreciated the valuable comments from the respected reviewer. We have corrected the errors in the revised manuscript and highlighted them.
According to the plagiarism analysis, 32% is detected. It is recommended to work on the manuscript to reduce this value.
Answer:
Thanks a lot for the valuable comment from the respected reviewer. In the revised version, we reduced this value to 5% using iThenticate. The iThenticate report was presented as an attached file.
There is a lack of discussion in the document, only results are discussed but there is no comparison or discussion with other works.
Answer:
Thanks a lot for the valuable suggestion from the respected reviewer. We added comparisons and discussions with other works and highlighted them within the revised manuscript.

Reviewer 2 Report
Dear Authors
Regarding to the Manuscript ID: food-2555100
Title: New healthy low-sugar and carotenoid-enriched-antioxidant beverage: Study of optimization and physicochemical properties
- The manuscript scientifically good and the experimental design
is appropriate to the subject and the results given are sufficient
Specific comments:
- The main question addressed by the research and found more results comparing with the previous
- The Abstract is good and contain all the need information
- The introduction and background are reasonable and given the premise of the paper
- The methodology is appreciate and was written in details.
- Some corrections of English and formatting are suggested
- The topic is original and relevant to the studying field
- The current study is added more information to the subject area compared with other published as it is suggest that the healthy sugar-free beverage with acceptable and promising sensorial, physiochemical, potential bioactive compounds, and antioxidant traits can have potential applications in enhancing health benefits and therapeutic purposes. .
- This paper is written in a clear style, and can be easily understood by readers.
- The conclusions consistent with the evidence and arguments presented.
- The study contain an appropriate and updated references
- The Tables and figures are comprehensive and helpful to understand all the studied parts.
- The results and discussion are sufficient and adequate
- The authors should do not start sentence with abbreviation, it is better to start with complete form, correct all over the manuscript.
- The authors should do not start sentence with a number
- There are several comments provided in the attached manuscript should be taken in consideration.
- Concerning to the above mentioned weaknesses would improve the overall quality and readability of the manuscript, enhancing its value for scholarly communication and further publication.
Best regards

This paper is written in a clear style, and can be easily understood by readers but, Some corrections of English and formatting are suggested
Author Response
The manuscript scientifically good and the experimental design
is appropriate to the subject and the results given are sufficient
Thank you for the positive feedback and your helpful comments and suggestions on the manuscript. We considered these comments, and we revised our manuscript according to the comments as follows:
Specific comments:
- The main question addressed by the research and found more results comparing with the previous
- The Abstract is good and contain all the need information
- The introduction and background are reasonable and given the premise of the paper
- The methodology is appreciate and was written in details.
- Some corrections of English and formatting are suggested
- The topic is original and relevant to the studying field
- The current study is added more information to the subject area compared with other published as it is suggest that the healthy sugar-free beverage with acceptable and promising sensorial, physiochemical, potential bioactive compounds, and antioxidant traits can have potential applications in enhancing health benefits and therapeutic purposes.
- This paper is written in a clear style, and can be easily understood by readers.
- The conclusions consistent with the evidence and arguments presented.
- The study contain an appropriate and updated references
- The Tables and figures are comprehensive and helpful to understand all the studied parts.
- The results and discussion are sufficient and adequate
- The authors should do not start sentence with abbreviation, it is better to start with complete form, correct all over the manuscript.
- The authors should do not start sentence with a number
- There are several comments provided in the attached manuscript should be taken in consideration.
- Concerning to the above mentioned weaknesses would improve the overall quality and readability of the manuscript, enhancing its value for scholarly communication and further publication.
Best regards
Answer:
We appreciate the valuable comments and suggestions from the respected reviewer and are very delighted with the positive feedback. We have considered all the valuable comments and suggestions from the respected reviewer. Furthermore, we have carefully revised the whole manuscript and made English and formatting corrections. The errors in the revised manuscript were also corrected.

Reviewer 3 Report
The study is of high interest for the industrials. Given the fact that it is an international collaboration, how can you expect your findings might get to the industrials? It is only my curiosity. Have you thought about some in vitro (gastrointestinal simulations to prove the bioavailability) or even in vivo (clinical) studies in the near future? Commonly, as food scientists, we are not able to claim a product as functional if there is no evidence of it, such as those experiments mentioned above. Based on your current findings, you can call it ``potentially functional``. The mention of the functional properties of each ingredient based on previously published literature is not enough.
Some text editing is necessary as there are different fonts used in the manuscript. Also, the manuscript structure imposed by the journal is not respected. Please check carefully.
Use abbreviations only if you had previously defined in the manuscript.
Some sentences in the same paragraph have no connection to each other.
Not always the antioxidant activity is of great help for the human body. Please state the side effects of these antioxidant properties enhanced products.
At the moment the study is impossible to be replicated. No evidence is given regarding the raw materials used in the study. What is their composition? You have to offer a clear composition description/characterization for each of them.
Please indicate the abbreviations below each table.
Results and Discussion and Materials and Methods sections are mixed. Please restructure them separately.
No details are given regarding the sensory analysis.
The English language style must be improved. There were encountered some errors concerning spelling and grammar. Please check carefully the entire manuscript.
Author Response
The study is of high interest for the industrials. Given the fact that it is an international collaboration, how can you expect your findings might get to the industrials? It is only my curiosity. Have you thought about some in vitro (gastrointestinal simulations to prove the bioavailability) or even in vivo (clinical) studies in the near future? Commonly, as food scientists, we are not able to claim a product as functional if there is no evidence of it, such as those experiments mentioned above. Based on your current findings, you can call it ``potentially functional``. The mention of the functional properties of each ingredient based on previously published literature is not enough.
Thank you for the positive feedback and your valuable comments on our work. We appreciate all the valuable suggestions.
We really appreciate the valuable comment from the respected reviewer. The authors agree with the reviewer’s suggestion. Some in vitro (gastrointestinal simulations to prove the bioavailability) or even in vivo (clinical) studies are needed in the near future to claim that this healthy sugar-free beverage is functional. After performing the mentioned experiments, this developed beverage can have potential applications and a good potential for commercialization as a functional beverage. Accordingly, instead of the term "functional," "potentially functional" was substituted in the text. Thus, in vitro or in vivo studies are a potentially important field for future studies. In the continuation of this comment, related explanations were added and highlighted within the manuscript in the conclusions section.
We carefully revised our manuscript according to the comments as follows:
Some text editing is necessary as there are different fonts used in the manuscript. Also, the manuscript structure imposed by the journal is not respected. Please check carefully.
Answer:
Thanks a lot for the valuable comment from the respected reviewer. We edited the whole manuscript carefully.
Use abbreviations only if you had previously defined in the manuscript. Some sentences in the same paragraph have no connection to each other.
Answer:
Thanks a lot for the valuable comment from the respected reviewer. We edited the whole manuscript carefully.
Not always the antioxidant activity is of great help for the human body. Please state the side effects of these antioxidant properties enhanced products.
Answer:
We appreciated the valuable comment from the respected reviewer. Following the dear reviewer's suggestion, we added some explanations in the Discussion section about the side effects of these products and highlighted them.
At the moment the study is impossible to be replicated. No evidence is given regarding the raw materials used in the study. What is their composition? You have to offer a clear composition description/characterization for each of them.
Answer:
Thanks a lot for the valuable comment from the respected reviewer. The composition description/characterization of the raw materials were added and highlighted within the revised manuscript.
Please indicate the abbreviations below each table.
Answer:
Thanks a lot for the valuable comment from the respected reviewer. In the continuation of this comment, the abbreviations below each table were indicated and highlighted within the revised manuscript.
Results and Discussion and Materials and Methods sections are mixed. Please restructure them separately.
Answer:
According to the dear reviewer's comment, we have restructured the Results and Discussion and Materials and Methods sections separately.
No details are given regarding the sensory analysis.
Answer:
Thanks a lot for the valuable suggestion from the respected reviewer. Following the dear reviewer's suggestion, we have added and highlighted more details regarding the sensory analysis in the revised version in Section 3.3.
Comments on the Quality of English Language
The English language style must be improved. There were encountered some errors concerning spelling and grammar. Please check carefully the entire manuscript.
Answer:
Thanks a lot for the valuable comment from the respected reviewer. We have revised the whole manuscript carefully and tried to avoid any spelling or grammar errors.
